# Joint Extraction of Entities and Relations Based on a Novel Tagging Scheme

## Abstract

Joint extraction of entities and relations is an important task in information extraction. To tackle this problem, we firstly propose a novel tagging scheme that can convert the joint extraction task to a tagging problem. Then, based on our tagging scheme, we study different end-to-end models to extract entities and their relations directly, without identifying entities and relations separately. We conduct experiments on a public dataset produced by distant supervision method and the experimental results show that the tagging based methods are better than most of the existing pipelined and joint learning methods. What's more, the end-to-end model proposed in this paper, achieves the best results on the public dataset.

## 1 Introduction

Joint extraction of entities and relations is to detect entity mentions and recognize their semantic relations simultaneously from unstructured text, as Figure 1 shows. Different from open information extraction (Open IE) (Banko et al., 2007) whose relation words are extracted from the given sentence, in this task, relation words are extracted from a predefined relation set which may not appear in the given sentence. It is an important issue in knowledge extraction and automatic construction of knowledge base.

Traditional methods handle this task in a pipelined manner, i.e., extracting the entities (Nadeau and Sekine, 2007) first and then recognizing their relations (Rink, 2010). This separated framework makes the task easy to deal with, and each component can be more flexible. But it neglects the relevance between these two sub-tasks

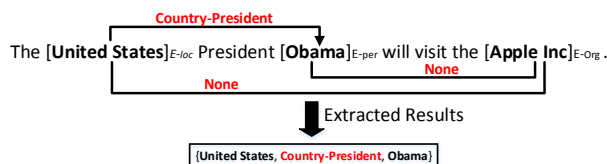

Figure 1: A standard example sentence for the task. "Country-President" is a relation in the predefined relation set.

and each subtask is an independent model. The results of entity recognition may affect the performance of relation classification and lead to erroneous delivery (Li and Ji, 2014).

Different from the pipelined methods, joint learning framework is to extract entities together with relations using a single model. It can effectively integrate the information of entities and relations, and it has been shown to achieve better results in this task. However, most existing joint methods are feature-based structured systems (Li and Ji, 2014; Miwa and Sasaki, 2014; Yu and Lam, 2010; Ren et al., 2017). They need complicated feature engineering and heavily rely on the other NLP toolkits, which might also lead to error propagation. In order to reduce the manual work in feature extraction, recently, (Miwa and Bansal, 2016) presents a neural network-based method for the end-to-end entities and relations extraction. Although the joint models can represent both entities and relations with shared parameters in a single model, they also extract the entities and relations separately and produce redundant information. For instance, the sentence in Figure 1 contains three entities: "United States", "Obama" and "Apple Inc". But only "United States" and "Obama" hold a fix relation "Country-President". Entity "Apple Inc" has no obvious relationship with the other entities in this sentence. Hence, the extracted result from this sentence is {**United States**$_{e1}$, **Country-President**$_r$,

Obama$_{e2}$}, which called triplet here.

In this paper, we focus on the extraction of triplets that are composed of two entities and one relation between these two entities. Therefore, we can model the triplets directly, rather than extracting the entities and relations separately. Based on the motivations, we propose a tagging scheme accompanied with the end-to-end model to settle this problem. We design a kind of novel tags which contain the information of entities and the relationships they hold. Based on this tagging scheme, the joint extraction of entities and relations can be transformed into a tagging problem. In this way, we can also easily use neural networks to model the task without complicated feature engineering.

Recently, end-to-end models based on LSTM (Hochreiter and Schmidhuber, 1997) have been successfully applied to various tagging tasks: Named Entity Recognition (Lample et al., 2016), CCG Supertagging (Vaswani et al., 2016), Chunking (Zhai et al., 2017) et al. LSTM is capable of learning long-term dependencies, which is beneficial to sequence modeling tasks. Therefore, based on our tagging scheme, we investigate different kinds of LSTM-based end-to-end models to jointly extract the entities and relations. We also modify the decoding method by adding a bias loss to make it more suitable for our special tags.

The method we proposed is a supervised learning algorithm. In reality, however, the process of manually labeling a training set with a large number of entity and relation is too expensive and error-prone. Therefore, we conduct experiments on a public dataset[1] which is produced by distant supervision method (Ren et al., 2017) to validate our approach. The experimental results show that our tagging scheme is effective in this task. In addition, our end-to-end model can achieve the best results on the public dataset.

The major contributions of this paper are: (1) A novel tagging scheme is proposed to jointly extract entities and relations, which can easily transform the extraction problem into a tagging task. It is the first work to solve the problem by a tagging manner. (2) Based on our tagging scheme, we study different kinds of end-to-end models to settle the problem. The tagging-based methods are better than most of the existing pipelined and joint learning methods. (3) Furthermore, we also develop an end-to-end model with a bias objective

[1]https://github.com/shanzhenren/CoType

function to suit for the novel tags. It can enhance the association between related entities.

## 2 Related Works

Entities and relations extraction is an important step to construct a knowledge base, which can be benefit for many NLP tasks (Zou et al., 2014). Two main frameworks have been widely used to solve the problem of extracting entity and their relationships. One is the pipelined method and the other is the joint learning method. The pipelined method treats this task as two separated tasks, i.e., named entity recognition (NER) (Nadeau and Sekine, 2007) and relation classification (RC) (Rink, 2010). Classical NER models are linear statistical models, such as Hidden Markov Models (HMM) and Conditional Random Fields (CRF) (Passos et al., 2014; Luo et al., 2015). Recently, several neural network architectures (Chiu and Nichols, 2015; Huang et al., 2015; Lample et al., 2016) have been successfully applied to NER, which is regarded as a sequential token tagging task. Existing methods for relation classification can also be divided into handcrafted feature based methods (Rink, 2010; Kambhatla, 2004) and neural network based methods (Xu, 2015a; Zheng et al., 2016; Zeng, 2014; Xu, 2015b; dos Santos, 2015). While joint models extract entities and relations using a single model. Most of the joint methods are feature-based structured systems (Ren et al., 2017; Yang and Cardie, 2013; Singh et al., 2013; Miwa and Sasaki, 2014; Li and Ji, 2014). Recently, (Miwa and Bansal, 2016) uses a LSTM-based model to extract entities and relations, which can reduce the manual work.

Different from the above methods, the method proposed in this paper is based on a special tagging manner, so that we can easily use end-to-end model to extract results without NER and RC. end-to-end method is to map the input sentence into meaningful vectors and then back to produce a sequence. It is widely used in machine translation (Kalchbrenner and Blunsom, 2013; Sutskever et al., 2014) and sequence tagging tasks (Lample et al., 2016; Vaswani et al., 2016). Most methods apply bidirectional LSTM to encode the input sentences, but the decoding methods are always different. For examples, (Lample et al., 2016) use a CRF layers to decode the tag sequence, while (Vaswani et al., 2016; Katiyar and Cardie, 2016) apply LSTM layer to produce the tag sequence.

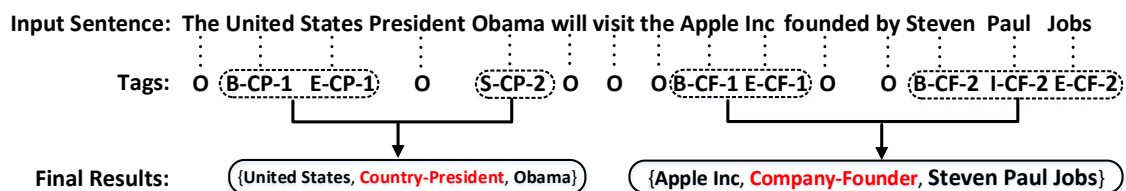

Figure 2: Gold standard annotation for an example sentence based on our tagging scheme, where "CP" is short for "Country-President" and "CF" is short for "Company-Founder".

## 3 Method

We propose a novel tagging scheme and an end-to-end model with biased objective function to jointly extract entities and their relations. In this section, we firstly introduce how to change the extraction problem to a tagging problem based on our tagging method. Then we detail the model we used to extract results.

### 3.1 The Tagging Scheme

Figure 2 is an example of how the results are tagged. Each word is assigned a label that contributes to extract the results. Tag "O" represents the "Other" tag, which means that the corresponding word is independent of the extracted results. In addition to "O", the other tags consist of three parts: the word position in the entity, the relation type, and the relation role. We use the "BIES" (Begin, Inside, End,Single) signs to represent the position information of a word in the entity. The relation type information is obtained from a predefined set of relations and the relation role information is represented by the numbers "1" and "2". An extracted result is represented by a triplet: $(Entity_1, RelationType, Entity_2)$. "1" means that the word belongs to the first entity in the triplet, while "2" belongs to second entity that behind the relation type. Thus, the total number of tags is $N_t = 2 * 4 * |R| + 1$, where $|R|$ is the size of the predefined relation set.

Figure 2 is an example illustrating our tagging method. The input sentence contains two triplets: {**United States, Country-President, Obama**} and {**Apple Inc, Company-Founder, Steven Paul Jobs**}, where "Country-President" and "Company-Founder" are the predefined relation types. The words "United","States","Obama","Apple","Inc" ,"Steven", "Paul" and "Jobs" are all related to the final extracted results. Thus they are tagged based

on our special tags. For example, the word of "United" is the first word of entity "United States" and is related to the relation "Country-President", so its tag is "B-CP-1". The other entity "Obama", which is corresponding to "United States", is labeled as "S-CP-2". Besides, the other words irrelevant to the final result are labeled as "O".

### 3.2 From Tag Sequence To Extracted Results

From the tag sequence in Figure 2, we know that "Obama" and "United States" share the same relation type "Country-President", "Apple Inc" and "Steven Paul Jobs" share the same relation type "Company-Founder". We combine entities with the same relation type into a triplet to get the final result. Accordingly, "Obama" and "United States" can be combined into a triplet whose relation type is "Country-President". Because, the relation role of "Obama" is "2" and "United States" is "1", the final result is {**United States, Country-President, Obama**}. The same applies to {**Apple Inc, Company-Founder, Steven Paul Jobs**}.

Besides, if a sentence contains two or more triplets with the same relation type, we combine every two entities into a triplet based on the nearest principle. For example, if the relation type "Country-President" in Figure 2 is "Company-Founder", then there will be four entities in the given sentence with the same relation type. "United States" is closest to entity "Obama" and the "Apple Inc" is closest to "Jobs", so the results will be {**United States, Company-Founder, Obama**} and {**Apple Inc, Company-Founder, Steven Paul Jobs**}. In this paper, we only consider the situation where an entity belongs to a triplet, and we leave identification of overlapping relations for future work.

### 3.3 The end-to-end Model

In recent years, end-to-end model based on neural network is been widely used in sequence tagging

task. In this paper, we investigate an end-to-end Model to produce the tags sequence. It contains a bi-directional Long Short Term Memory (Bi-LSTM) layer to encode the input sentence and a LSTM decoding layer with bias loss. The bias loss can enhance the relevance of entity tags.

**The Bi-LSTM Encoding Layer.** In sequence tagging problems, the Bi-LSTM encoding layer has been shown the effectiveness to capture the semantic information of each word. It contains word embedding layer, forward lstm layer, backward lstm layer and the concatenate layer. The word embedding layer converts the word with 1-hot representation to an embedding vector. Hence, a sequence of words can be represented as $W = \{w_1, ...w_t, w_{t+1}...w_n\}$, where $w_t \in \mathbb{R}^d$ is the $d$-dimensional word vector corresponding to the $t$-th word in the sentence and $n$ is the length of the given sentence. After word embedding layer, there are two parallel LSTM layers: forward lstm layer and backward lstm layer. For each word $w_t$, the forward layer will encode $w_t$ by considering the contextual information from word $w_1$ to $w_t$, which is marked as $\overrightarrow{h_t}$. In the similar way, the backward layer will encode $w_t$ based on the contextual information from $w_n$ to $w_t$, which is marked as $\overleftarrow{h_t}$.

The LSTM architecture consists of a set of recurrently connected subnets, known as memory blocks. Each time-step is a LSTM memory block, which is used to compute current hidden vector $h_t$ based on the previous hidden vector $h_{t-1}$, the previous cell vector $c_{t-1}$ and the current input word embedding $w_t$. It can be shortly denoted as: $\overrightarrow{h_t} = lstm(\overrightarrow{h_{t-1}}, \overrightarrow{c_{t-1}}, w_t)$ and $\overleftarrow{h_t} = lstm(\overleftarrow{h_{t+1}}, \overleftarrow{c_{t+1}}, w_t)$. The detail operations are defined as follows:

$$i_t = \delta(W_{xi}x_t + W_{hi}h_{t-1} + W_{ci}c_{t-1} + b_i), \quad (1)$$

$$f_t = \delta(W_{xf}x_t + W_{hf}h_{t-1} + W_{cf}c_{t-1} + b_f), \quad (2)$$

$$z_t = tanh(W_{xc}x_t + W_{hc}h_{t-1} + b_c), \quad (3)$$

$$c_t = f_t c_{t-1} + i_t z_t, \quad (4)$$

$$o_t = \delta(W_{xo}x_t + W_{ho}h_{t-1} + W_{co}c_t + b_o), \quad (5)$$

$$h_t = o_t tanh(c_t), \quad (6)$$

where $i$, $f$ and $o$ are the input gate, forget gate and output gate respectively, b is the bias term, c is the cell memory, $\cdot$ denotes element-wise multiplication and $W_{(.)}$ are the parameters. Finally, we concatenate $\overleftarrow{h_t}$ and $\overrightarrow{h_t}$ to represent word $t$'s encoding information, denoted as $h_t = [\overleftarrow{h_t}, \overrightarrow{h_t}]$.

**The LSTM Decoding Layer.** We also adopt a L-STM structure to produce the tag sequence. When detecting the tag of word $w_t$, the inputs of decoding layer are: $h_t$ obtained from Bi-LSTM encoding layer, former predicted tag vector $T_{t-1}$, and the former hidden state of decoding LSTM $s_{t-1}$. Each unit of the decoding LSTM is the same as the encoding lstm memory block except for the input gate, which can be rewritten as:

$$i_t = \delta(W_{xi}h_t + W_{hi}s_{t-1} + W_{ti}T_{t-1} + b_i), \quad (7)$$

where the tag embedding $T$ is transformed from the hidden state $s$ as follows:

$$T_t = W_{ts}s_t + b_{ts}. \quad (8)$$

The final softmax layer computes normalized entity tag probabilities based on the tag predicted vector $T_t$:

$$y_t = W_y T_t + b_y, \quad (9)$$

$$p_t^i = \frac{exp(y_t^i)}{\sum_{j=1}^{N_t} exp(y_t^j)}, \quad (10)$$

where $W_y$ is the softmax matrix, $N_t$ is the total number of tags. Because the $T$ is similar to tag embedding and LSTM is capable of learning long-term dependencies, the decoding manner can model tag interactions.

**The Bias Objective Function.** We train our model to maximize the log-likelihood of the data and the optimization method we used is RMSprop proposed by Hinton in (Tieleman and Hinton, 2012). The objective function can be defined as:

$$L = max \sum_{j=1}^{|\mathbb{D}|} \sum_{t=1}^{L_j} (log(p_t^{(j)} = y_t^{(j)}|x_j, \Theta) \cdot I(O)$$

$$+ \alpha \cdot log(p_t^{(j)} = y_t^{(j)}|x_j, \Theta) \cdot (1 - I(O))),$$

where $|\mathbb{D}|$ is the size of training set, $L_j$ is the length of sentence $x_j$, $y_t^{(j)}$ is the label of word $t$ in sentence $x_j$ and $p_t^{(j)}$ is the normalized probabilities of tags which defined in Formula 10. Besides, $I(O)$ is a switching function to distinguish the loss of tag 'O' and relational tags that can indicate the results. It is defined as follows:

$$I(O) = \begin{cases} 1, & if \quad tag = 'O' \\ 0, & if \quad tag \neq 'O'. \end{cases}$$

$\alpha$ is the bias weight. The larger $\alpha$ is, the greater influence of relational tags on the model.

## 4  Experiments

### 4.1  Experimental setting

**Dataset** To evaluate the performance of our methods, we use the public dataset NYT [2] which is produced by distant supervision method (Ren et al., 2017). A large amount of training data can be obtained by means of distant supervision methods without manually labeling. While the test set is manually labeled to ensure its quality. In total, the training data contains $1.18M$ sentences samples from New York Times News with $353k$ triplets. The test set contains 395 manually labeling sentences with $3,880$ triplets. Besides, the size of relation set is 24.

**Evaluation** We adopt standard Precision (Prec), Recall (Rec) and F1 score to evaluate the results. A triplet is regarded as correct when its relation type and the head offsets of two corresponding entities are both correct. Besides, the ground-truth relation mentions are given and "None" label is excluded as (Ren et al., 2017; Li and Ji, 2014; Miwa and Bansal, 2016) did. We create a validation set by randomly sampling $10\%$ data from test set and use the remaining data as evaluation based on (Ren et al., 2017)'s suggestion. We run 10 times for each experiment then report the average results and their standard deviation as Table 1 shows.

**Hyperparameters** Our model consists of a Bi-LSTM encoding layer and a LSTM decoding layer with bias objective function. The word embeddings used in the encoding part are initialed by running word2vec[3] (Mikolov et al., 2013) on NYT training corpus. The dimension of the word embeddings is $d = 300$. We regularize our network using dropout on embedding layer and the dropout ratio is $0.5$. The number of lstm units in encoding layer is 300 and the number in decoding layer is 600. The bias parameter $\alpha$ corresponding to the results in Table 1 is 10.

**Baselines** We compare our method with several classical triplet extraction methods, which can be divided into the following categories: the pipelined methods, the jointly extracting method-

---

[2]The dataset can be downloaded at: https://github.com/shanzhenren/CoType. There are three data sets in the public resource and we only use the NYT dataset. Because more than 50% of the data in BioInfer has overlapping relations which is beyond the scope of this paper. As for dataset Wiki-KBP, the number of relation type in the test set is more than that of the train set, which is also not suitable for a supervised training method.

[3]https://code.google.com/archive/p/word2vec/

s and the end-to-end methods based our tagging scheme.

For the pipelined methods, we follow (Ren et al., 2017)'s settings: The NER results are obtained by CoType (Ren et al., 2017) then several classical relation classification methods are applied to detect the relations. These methods are: (1) DS-logistic (Mintz et al., 2009) is a distant supervised and feature based method, which combines the advantages of supervised IE and unsupervised IE features; (2) LINE (Tang et al., 2015) is a network embedding method, which is suitable for arbitrary types of information networks; (3) FCM (Gormley et al., 2015) is a compositional model that combines lexicalized linguistic context and word embeddings for relation extraction.

The jointly extracting methods used in this paper are listed as follows: (4) DS-Joint (Li and Ji, 2014) is a supervised method, which jointly extracts entities and relations using structured perceptron on human-annotated dataset; (5) MultiR (Hoffmann et al., 2011) is a typical distant supervised method based on multi-instance learning algorithms to combat the noisy training data; (6) CoType (Ren et al., 2017) is a domain independent framework by jointly embedding entity mentions, relation mentions, text features and type labels into meaningful representations.

In addition, we also compare our method with two classical end-to-end tagging models: LSTM-CRF (Lample et al., 2016) and LSTM-LSTM (Vaswani et al., 2016). LSTM-CRF is proposed for entity recognition by using a bidirectional LSTM to encode input sentence and a conditional random fields to predict the entity tag sequence. Different from LSTM-CRF, LSTM-LSTM uses a LSTM layer to decode the tag sequence instead of CRF. They are used for the first time to jointly extract entities and relations based on our tagging scheme.

### 4.2  Experimental Results

We report the results of different methods as shown in Table 1. It can be seen that our method, LSTM-LSTM-Bias, outperforms all other methods in F1 score and achieves a $3\%$ improvement in $F1$ over the best method CoType (Ren et al., 2017). It shows the effectiveness of our proposed method. Furthermore, from Table 1, we also can see that the jointly extracting methods are better than pipelined methods, and the tagging methods

| Methods | *Prec.* | *Rec.* | *F1* |
|---|---|---|---|
| FCM | 0.553 | 0.154 | 0.240 |
| DS+logistic | 0.258 | 0.393 | 0.311 |
| LINE | 0.335 | 0.329 | 0.332 |
| MultiR | 0.338 | 0.327 | 0.333 |
| DS-Joint | 0.574 | 0.256 | 0.354 |
| CoType | 0.423 | **0.511** | 0.463 |
| LSTM-CRF | **0.693** ± **0.008** | 0.310 ± 0.007 | 0.428 ± 0.008 |
| LSTM-LSTM | 0.682 ± 0.007 | 0.320 ± 0.006 | 0.436 ± 0.006 |
| **LSTM-LSTM-Bias** | 0.615 ± 0.008 | 0.414 ± 0.005 | **0.495** ± **0.006** |

Table 1: The predicted results of different methods on extracting both entities and their relations. The first part (from row 1 to row 3) is the pipelined methods and the second part (row 4 to 6) is the jointly extracting methods. Our tagging methods are shown in part three (row 7 to 9). In this part, we not only report the results of precision, recall and F1, we also compute their standard deviation.

| Elements | E1 | | | E2 | | | (E1,E2) | | |
|---|---|---|---|---|---|---|---|---|---|
| PRF | *Prec.* | *Rec.* | *F1* | *Prec.* | *Rec.* | *F1* | *Prec.* | *Rec.* | *F1* |
| LSTM-CRF | **0.596** | 0.325 | 0.420 | 0.605 | 0.325 | 0.423 | **0.724** | 0.341 | 0.465 |
| LSTM-LSTM | 0.593 | 0.342 | 0.434 | **0.619** | 0.334 | 0.434 | 0.705 | 0.340 | 0.458 |
| LSTM-LSTM-Bias | 0.590 | **0.479** | **0.529** | 0.597 | **0.451** | **0.514** | 0.645 | **0.437** | **0.520** |

Table 2: The predicted results of triplet's elements based on our tagging scheme.

are better than most of the jointly extracting methods. It also validates the validity of our tagging scheme for the task of jointly extracting entities and relations.

When compared with the traditional methods, the precisions of the end-to-end models are significantly improved. But only LSTM-LSTM-Bias can be better to balance the precision and recall. The reason may be that these end-to-end models all use a Bi-LSTM encoding input sentence and different neural networks to decode the results. The methods based on neural networks can well fit the data. Therefore, they can learn the common features of the training set well and may lead to the lower expansibility. We also find that the LSTM-LSTM model is better than LSTM-CRF model based on our tagging scheme. Because, LSTM is capable of learning long-term dependencies and CRF (Lafferty et al., 2001) is good at capturing the joint probability of the entire sequence of labels. The related tags may have a long distance from each other. Hence, LSTM decoding manner is a little better than CRF. LSTM-LSTM-Bias adds a bias weight to enhance the effect of entity tags and weaken the effect of invalid tag. Therefore, in this tagging scheme, our method can be better than the common LSTM-decoding methods.

## 5 Analysis and Discussion

### 5.1 Error Analysis

In this paper, we focus on extracting triplets composed of two entities and a relation. Table 1 has shown the predict results of the task. It treats an triplet is correct only when the relation type and the head offsets of two corresponding entities are both correct. In order to find out the factors that affect the results of end-to-end models, we analyze the performance on predicting each element in the triplet as Table 2 shows. $E1$ and $E2$ represent the performance on predicting each entity, respectively. If the head offset of the first entity is correct, then the instance of $E1$ is correct, the same to $E2$. Regardless of relation type, if the head offsets of two corresponding entities are both correct, the instance of $(E1, E2)$ is correct.

As shown in Table 2, $(E1, E2)$ has higher precision when compared with $E1$ and $E2$. But its recall result is lower than $E1$ and $E2$. It means that some of the predicted entities do not form a pair. They only obtain $E1$ and do not find its corresponding $E2$, or obtain $E2$ and do not find its corresponding $E1$. Thus it leads to the prediction of more single $E$ and less $(E1, E2)$ pairs. Therefore, entity pair $(E1, E2)$ has higher precision and lower recall than single $E$. Besides, the predict-

ed results of $(E1, E2)$ in Table 2 have about 3% improvement when compared predicted results in Table 1, which means that 3% of the test data is predicted to be wrong because the relation type is predicted to be wrong.

## 5.2 Analysis of Bias

Different from LSTM-CRF and LSTM-LSTM, our approach is biased towards relational labels to enhance links between entities. In order to further analyze the effect of the bias objective function, we visualize the ratio of predicted single entities for each end-to-end method as Figure 3. The s-ingle entities refer to those who cannot find their corresponding entities. Figure 3 shows whether it is E1 or E2, our method can get a relatively low ra-tio on the single entities. It means that our method can effectively associate two entities when com-pared LSTM-CRF and LSTM-LSTM which pay little attention to the relational tags.

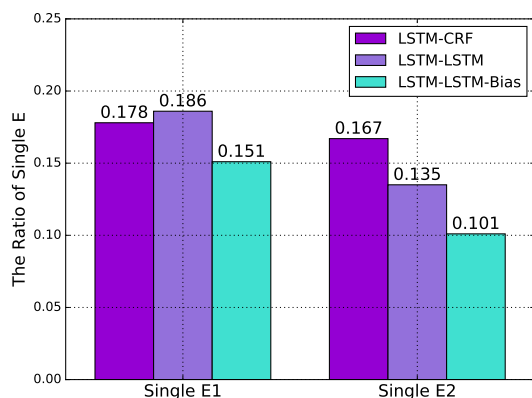

Figure 3: The ratio of predicted single entities for each method. The higher of the ratio the more en-tities are left.

Besides, we also change the Bias Parameter $\alpha$ from 1 to 20, and the predicted results are shown in Figure 4. If $\alpha$ is too large, it will affect the accuracy of prediction and if $\alpha$ is too small, the recall will decline. When $\alpha = 10$, LSTM-LSTM-Bias can balance the precision and recall, and can achieve the best F1 scores.

## 5.3 The Applicability of Our Methods

Although LSTM-LSTM-Bias can achieve a max-imum $F1$ value, in practice we may pay more at-tention to the precision or recall. Because, LSTM-LSTM and LSTM-LSTM-Bias use the softmax

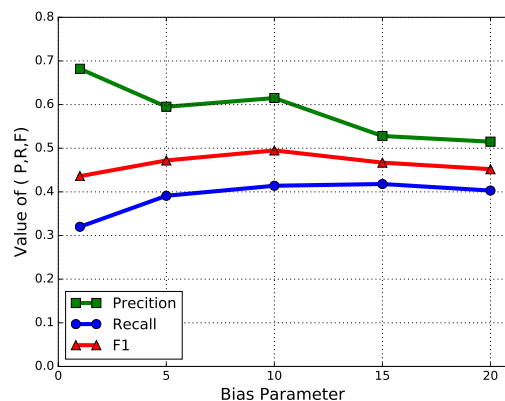

Figure 4: The results predicted by LSTM-LSTM-Bias on different bias parameter $\alpha$.

function to select the maximum probability label as the predicted tag for each word. We reset the prediction method as follows: if the maximum predicted probability is smaller than the threshold, the predicted tag is 'O', else the result is the max-imum probability label. Therefore, we tune the threshold for LSTM-LSTM and LSTM-LSTM-Bias to obtain their Precision-Recall curves that can reflect the predictive ability of models. Fig-ure 5 shows that LSTM-LSTM-Bias is larger than LSTM-LSTM in both precision and recall, which means a more broad applicability.

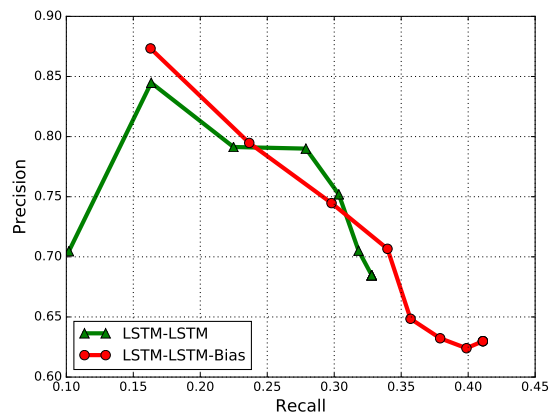

Figure 5: Precision-Recall curves of LSTM-LSTM and LSTM-LSTM-Bias in jointly extract-ing entities and relation task.

## 5.4 Case Study

In this section, we observe the prediction results of end-to-end methods, and then select several repre-sentative examples to illustrate the advantages and

| Standard S1 | [Panama City Beach]$_{E2contain}$ has condos , but the area was one of only two in [Florida]$_{E1contain}$ where sales rose in March , compared with a year earlier. |
|---|---|
| LSTM-LSTM | Panama City Beach has condos , but the area was one of only two in [Florida]$_{E1contain}$ where sales rose in March , compared with a year earlier. |
| LSTM-LSTM-Bias | [Panama City Beach]$_{E2contain}$ has condos , but the area was one of only two in [Florida]$_{E1contain}$ where sales rose in March , compared with a year earlier. |
| Standard S2 | All came from [Nuremberg]$_{E2contain}$ , [Germany]$_{E1contain}$ , a center of brass production since the Middle Ages. |
| LSTM-LSTM | All came from Nuremberg , [Germany]$_{E1contain}$ , a center of brass production since the [Middle Ages]$_{E2contain}$. |
| LSTM-LSTM-Bias | All came from Nuremberg , [Germany]$_{E1contain}$ , a center of brass production since the [Middle Ages]$_{E2contain}$. |
| Standard S3 | [Stephen A.]$_{E2CF}$ , the co-founder of the [Blackstone Group]$_{E1CF}$, which is in the process of going public , made $ 400 million last year. |
| LSTM-LSTM | [Stephen A.]$_{E1CF}$ , the co-founder of the [Blackstone Group]$_{E1CF}$, which is in the process of going public , made $ 400 million last year. |
| LSTM-LSTM-Bias | [Stephen A.]$_{E1CF}$ , the co-founder of the [Blackstone Group]$_{E2CF}$, which is in the process of going public , made $ 400 million last year. |

Table 3: Output from different models. Standard $S_i$ represents the gold standard of sentence $i$. The blue part is the correct result, and the red one is the wrong one. $E1CF$ in case '3' is short for $E1_{Company-Founder}$.

disadvantages of the methods. Each example contains three row, the first row is the gold standard, the second and the third rows are the extracted results of model LSTM-LSTM and LSTM-LSTM-Bias respectively. Example $S1$ represents the situation that the distance between the two interrelated entities is far away from each other, which is more difficult to detect their relationships. When compared with LSTM-LSTM, LSTM-LSTM-Bias uses a bias objective function which enhance the relevance between entities. Therefore, in this example, LSTM-LSTM-Bias can extract two related entities, while LSTM-LSTM can only extract one entity of "Florida" and can not detect entity "Panama City Beach".

$S2$ is a negative example that shows these methods may mistakenly predict one of the entity. There are no indicative words between entities $Nuremberg$ and $Germany$. Besides, the patten "a * of *" between $Germany$ and $MiddleAges$ may be easy to mislead the models that there exists a relation of "Contains" between them. The problem can be solved by adding some samples of this kind of expression patterns to the training data.

$S3$ is a case that models can predict the entities' head offset right, but the relational role is

wrong. LSTM-LSTM treats both "Stephen A. Schwarzman" and "Blackstone Group" as entity $E1$, and can not find its corresponding $E2$. Although, LSTM-LSMT–Bias can find the entities pair $(E1, E2)$, it reverses the roles of "Stephen A. Schwarzman" and "Blackstone Group". It shows that LSTM-LSTM-Bias is able to better on predicting entities pair, but it remains to be improved in distinguishing the relationship between the two entities.

## 6 Conclusion

In this paper, we propose a novel tagging scheme and investigate the end-to-end models to jointly extract entities and relations. The experimental results show the effectiveness of our proposed method. But it still has shortcoming on the identification of the overlapping relations. In the future work, we will replace the softmax function in the output layer with multiple classifier, so that a word can has multiple tags. In this way, a word can appear in multiple triplet results, which can solve the problem of overlapping relations. Although, our model can enhance the effect of entity tags, the association between two corresponding entities still requires refinement in next works.

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
