# Peer review of "Joint Extraction of Entities and Relations Based on a Novel Tagging Scheme"

_ACL 2017 — decision unknown_

[Official Review · Reviewer 1 · rating 5 · confidence 5]
soundness 3 · originality 3 · clarity 5 · impact 3 · substance 5 · appropriateness 5 · meaningful comparison 3 · presentation format Oral Presentation

- Strengths:

When introducing the task, the authors use illustrative examples as well as the
contributions of this paper. 
Related Works section covers the state of the art, at the same time pointing
similarities and differences between related Works and the proposed method.
The presentation of the method is very clear, since the authors separate the
tagging scheme and the end-to-end model.
Another strong point of this work is the baselines used to compare the proposed
methods with several classical triplet extraction methods.
At last, the presentation of examples from dataset used to illustrate the
advantages and disadvantages of the methods was very important. These outputs
complement the explanation of tagging and evaluation of triplets. 

- Weaknesses:

One of the main contributions of this paper is a new tagging scheme described
in Section 3.1, however there are already other schemes for NER and RE being
used, such as IO, BIO and BILOU. 
Did the authors perform any experiment using other tagging scheme for this
method?
Regarding the dataset, in line 14, page 5, the authors cite the number of
relations (24), but they do not mention the number or the type of named
entities.
In Section 4.1, the evaluation criteria of triplets are presented. These
criteria were based on previous work? As I see it, the stage of entity
identification is not complete if you consider only the head of the entity.
Regarding example S3, shown in Table 3, the output of the LSTM-LSTM-Bias was
considered correct? The text states that the relation role is wrong, although
it is not clear if the relation role is considered in the evaluation. 

- General Discussion:

This paper proposes a novel tagging scheme and investigates the end-to-end
models to jointly extract entities and relations. 
The article is organized in a clear way and it is well written, which makes it
easy to understand the proposed method.